# Use of artificial intelligence in the detection of primary prostate cancer in multiparametric MRI with its clinical outcomes: a protocol for a systematic review and meta-analysis

Maya Thomas [1], Sanjana Murali,[1] Benjamin Scott S Simpson [2], Alex Freeman,[3] Alex Kirkham,[4] Daniel Kelly [5], Hayley C Whitaker,[6] Yi Zhao [1], Mark Emberton [6,7], Joseph M Norris [6,7]

MT and SM are joint first authors.
ME and JMN are joint senior authors.

**Correspondence to**
Maya Thomas;
maya.thomas18@imperial.ac.uk

## ABSTRACT

**Introduction** Multiparametric MRI (mpMRI) has transformed the prostate cancer diagnostic pathway, allowing for improved risk stratification and more targeted subsequent management. However, concerns exist over the interobserver variability of images and the applicability of this model long term, especially considering the current shortage of radiologists and the growing ageing population. Artificial intelligence (AI) is being integrated into clinical practice to support diagnostic and therapeutic imaging analysis to overcome these concerns. The following report details a protocol for a systematic review and meta-analysis investigating the accuracy of AI in predicting primary prostate cancer on mpMRI.

**Methods and analysis** A systematic search will be performed using PubMed, MEDLINE, Embase and Cochrane databases. All relevant articles published between January 2016 and February 2023 will be eligible for inclusion. To be included, articles must use AI to study MRI prostate images to detect prostate cancer. All included articles will be in full-text, reporting original data and written in English. The protocol follows the Preferred Reporting Items for Systematic Review and Meta-Analysis Protocols 2015 checklist. The QUADAS-2 score will assess the quality and risk of bias across selected studies.

**Ethics and dissemination** Ethical approval will not be required for this systematic review. Findings will be disseminated through peer-reviewed publications and presentations at both national and international conferences.

**PROSPERO registration number** CRD42021293745.

## STRENGTHS AND LIMITATIONS OF THIS STUDY

⇒ This protocol is written in line with Preferred Reporting Items for Systematic Reviews and Meta-Analyses guidelines and will include both subgroup and sensitivity analyses to further investigate the heterogeneity of included studies.

⇒ Artificial intelligence (AI)-specific metrics such as the F1-score and precision-recall area under the curve will be employed to mitigate the restrictions of conventional pooled analysis, such as the impact of class imbalance.

⇒ As AI is a relatively novel technology in multiparametric MRI (mpMRI), the long-term data on clinical outcomes associated with its use may be limited.

⇒ The restrictions on language and use of mpMRI only may result in few suitable studies for inclusion.

unnecessary healthcare cost and detrimental effects on patient mental health.[1–3]

The introduction of multiparametric MRI (mpMRI) was pivotal in resolving this issue. Multiple sources now recommend using mpMRI as a prebiopsy investigation[2–7] to allow earlier risk stratification of those with suspected prostate cancer and improve the utility of the biopsies performed by enabling a more targeted approach. Despite this, interobserver variability of images remains a major concern,[5 8] with a lack of standardised protocols and a requirement for experienced radiologists being examples of contributory factors.[4 9] While radiology trainees show rapid improvement in the accuracy of interpreting studies with increased exposure, there is still a margin of error at around 25%.[10] Additionally, prostate cancer is the second most frequent cancer diagnosis globally, and the healthcare burden is only set to increase due to a growing ageing population. Therefore,

## BACKGROUND

Accurately identifying clinically significant prostate cancer has been a long-standing, challenging issue. Previous diagnostic pathways have resulted in many patients undergoing biopsies being diagnosed with clinically insignificant prostate cancer. This can lead to potentially aggressive overtreatment,

BMJ

the volume of diagnostic imaging and management monitoring required of radiologists will also likely increase over time.[11]

Artificial intelligence (AI), including deep learning (DL), describes how computers can be programmed to imitate human intelligence.[12] AI can be used in a semiautomated or fully automated approach. Semiautomated AI refers to using more traditional machine learning mechanisms such as radiomics, where radiologists must perform some preprocessing of images to make them algorithm appropriate. Fully automated AI has the benefit of using neural networks to perform feature identification.[13] Neural networks are a subset of DL models that mimic the human visual cortex.[14] Each neural network layer comprises neurons that identify several features of an image by applying multiple edge, colour and texture filters.[15] These neural networks are trained to associate specific characteristics with certain classifications by iterating through a training dataset of images. These trained layers can work together to perform feature identification on unseen photos; for example, in mpMRI, they could segment a potentially suspicious lesion in the prostate.[16 17] AI is being integrated into clinical practice to support diagnoses, assist in therapeutic decisions and predict patient outcomes.[18] Several studies have assessed the level of agreement between AI and DL to clinical radiologists to interpret mpMRI in prostate cancer investigation.[19–22] Application of AI could improve the current interobserver variability, expedite mpMRI interpretation[21] and risk-stratify potentially suspicious lesions for radiologist review.

However, it is also essential to address that DL algorithms are not yet advanced enough for unsupervised clinical usage.[19 23] Many issues surround the training of AI models, such as lack of diversity in training datasets—primarily in ethnicity, with many cohorts using predominantly white patients,[24] even though prostate cancer is more prevalent and aggressive in Afro-Caribbean men.[11] In many studies, there is a lack of external validation, which makes it unclear how models would perform on new data, and also runs the risk of overfitting to training data—where the model is excellent at recognising lesions in training data but is unable to recognise lesions in new data.[24] Class imbalance due to most negative cases can also present as an issue by creating algorithm bias and false negatives, which warrants review by senior radiologists.[24]

Furthermore, variations of the Prostate Imaging Reporting and Data System (PI-RADS) exist—with version 1 introduced in 2011, replaced by version 2 in 2016, and version 2.1 in 2019.[25 26] This makes evaluating how comparable the AI is more challenging due to a lack of consistency. This study will explore this in more detail to inform future reporting methodologies if different PI-RADS versions are used in the included studies. As some previous studies report heterogeneity among studies,[13 19 27] we hope to ascertain the level of heterogeneity in current studies and the impact of specific study methodologies on overall heterogeneity.

This systematic review and meta-analysis aims to summarise the accuracy of AI in predicting prostate cancer on mpMRI and how its use may impact clinical outcomes in patients.

## METHODS AND ANALYSIS

The protocol for this systematic review follows the Preferred Reporting Items for Systematic Review and Meta-Analysis Protocols 2015 checklist.[28] The study has been prospectively registered with PROSPERO review databases, and all methods described here were established before implementation. The statistical data evaluating the sensitivity, specificity, positive predictive value (PPV), negative predictive value (NPV) and area under the curve (AUC) of the use of AI in mpMRI prostate imaging for the detection of prostate cancer will be derived following a thorough analysis and thematic synthesis of included studies. These studies' pooled sensitivities and specificities will be determined before PPV and NPV values are derived.

### Search methodology

A systematic search will be performed using PubMed, MEDLINE, Embase and Cochrane databases. The search strategy will include using medical subject heading (MeSH) terms and free text with appropriate Boolean operators and will encompass articles from January 2016 to February 2023 to achieve maximum yield of the relevant evidence. The search will include the following key terms: "prostate", "cancer", "diagnosis" and multiple synonyms for "mpMRI", "artificial intelligence" and "deep learning". The full search strategy is detailed in online supplemental file S1. To facilitate the initial screening process, Rayyan will be employed, a semiautomated application designed to improve the speed and reporting accuracy of systematic reviews.[29] All eligible articles from the initial search will be uploaded to Rayyan. A further manual search of the references in all included articles will be performed to identify any further relevant literature not found by the initial search strategy. If data is absent or ambiguous, the corresponding authors will be contacted for clarification.

### Study selection and data extraction

The screening process will be completed independently by two researchers (MT and SM). Titles and abstracts of eligible studies will be assessed, and irrelevant articles will be removed. A full-text version of relevant articles will be downloaded for further eligibility review. Any dispute among researchers will be discussed, and a third reviewer (YZ) will be consulted, with the issue being resolved by consensus. The reasoning for excluding articles will be documented and detailed in a Preferred Reporting Items for Systematic Reviews and Meta-Analyses flow diagram. Before commencing the screening process, calibration exercises will be conducted to maintain consistency between the researchers, reducing inter-reviewer bias.

**Table 1** Data collection items

| Item No | Data title | Data type |
|---|---|---|
| 1 | Year of publication | Study characteristic |
| 2 | Study authors | Study characteristic |
| 3 | Experimental design | Study characteristic |
| 4 | Patient population | Demographics |
| 5 | Study size | Demographics |
| 6 | Type of MRI imaging used | Methodology |
| 7 | Type of MRI scoring system used | Methodology |
| 8 | Type of artificial intelligence models used | Methodology |
| 9 | Definition of clinically significant disease | Methodology |
| 10 | Predictive performances | Outcome |

### Inclusion and exclusion criteria

For inclusion in this systematic review, studies must use either fully automated or semiautomated AI to study mpMRI prostate images to detect prostate cancer. Prospective or retrospective studies can be included. Comparisons will be focused on evaluating sensitivities, specificities, PPVs, NPVs and AUCs. Patient cohorts can comprise suspected and confirmed incidences of cancer, and studies included should include reference standards using histological findings.

Correspondence papers, ongoing studies, case reports and conference abstracts will be excluded from this analysis. Articles which are not written in the English language will also be excluded. Papers not using mpMRI as the diagnostic modality will be excluded. Studies which include patients with previous treatment of prostate cancer will be excluded.

### Data extraction (table of collection)

The following data detailed in table 1 will be collected from all included studies. The researchers will perform data extraction independently, collating collected data in a dedicated datasheet. Any discrepancies in data extraction will be examined by a third reviewer and will be resolved by consensus.

If the data is available, the relevant figures will be extracted appropriately, including but not limited to true positive, true negative, false positive and false negatives and derivatives of these calculations. If these are not provided, then attempts will be made to calculate from the data provided. If this is unsuccessful, the relevant authors of the paper in question will be contacted to provide the data.

### Endpoints

The primary endpoint will be statistically significant quantified accuracy in using AI in MRI prostate imaging to detect prostate cancer and determine whether AI could be deployed in the clinical decision-making process. Additional outcomes will include other parameters investigating the extent of the disease.

### Meta-analysis

A meta-analysis will be conducted if there are a sufficient number of appropriate studies. This would aim to obtain a pooled quantitative diagnostic accuracy value for AI performance detecting prostate cancer from mpMRI. To begin, sensitivity and specificity values will be retrieved, or if studies do not provide these values, they will be calculated from clinical tables or requested from corresponding authors. If a considerable proportion of included studies used another metric, these values would be retrieved and calculated separately. This may include common AI-focused metrics such as F1-score and precision-recall AUC. The distributions of the untransformed, logit and double-arcsine transformed proportions will be compared and assessed for normality using density plots and Shapiro-Wilk tests, with whichever set of ratios most resembling normal distribution being used for further analysis.

Inter-study variation will be quantified through $I^2$, and if statistically significant, a random-effects model will be fitted for estimation of the summary estimate. Once the model fits all relevant studies, leave-one-out analysis (LOO) and accompanying diagnostic plots will be used to identify influential studies. This will include externally studentised residuals, the difference in fits values, Cook's distances, covariance ratios, LOO estimates of the amount of heterogeneity, LOO values of the test statistics for heterogeneity, hat values and weights. Each study will be removed one at a time, and the summary proportions re-estimated based on the remaining $n-1$ studies. Studies with a statistically significant influence on the fitted model will be considered outliers and removed. The model will then be re-fitted. A summary estimate comprising the remaining studies will then be calculated to estimate the accuracy using AI in mpMRI for prostate cancer detection. All data analysis and visualisation will be performed using the R statistical environment using the 'mvmeta' and 'meta' packages.

### Risk of bias in individual studies

The QUADAS-2 score will be employed to assess the quality and the risk of bias across the selected studies.[30] This tool comprises four domains: patient selection, index test, reference standard and flow and timing. The quality of the research methodology for each study will be assessed within these four domains. Two (MT and YZ) reviewers will be independently involved in this process, with any disparities resolved by consensus. The bias assessment will evaluate the applicability and reliability of the data used. This will inform evidence synthesis and increase transparency as articles may be excluded if found to be of low quality or suggestive of high levels of bias, or if included, appropriate commentaries will be incorporated into the discussion.

## Ethics and dissemination

As primary data is not being collected, no ethical approval is required for this study. The results of this systematic review will be published in a peer-reviewed academic journal and scientific conferences. Any collected data will be made available as supplemental material for full transparency.

## Patient and public involvement

There will be no patient or public involvement in this study.

## Trial status

► Preliminary searches: Started.
► Piloting of the study selection process: Started.
► Formal screening: Not started.
► Data extraction: Not started
► Risk of bias assessment: Not started.
► Data analysis: Not started.

**Author affiliations**
[1]School of Medicine, Imperial College London, London, UK
[2]UCL Cancer Institute, University College London, London, UK
[3]Department of Pathology, University College London Hospitals NHS Foundation Trust, London, UK
[4]Department of Radiology, University College London Hospitals NHS Foundation Trust, London, UK
[5]School of Healthcare Sciences, Cardiff University, Cardiff, UK
[6]UCL Division of Surgery and Interventional Science, University College London, London, UK
[7]Department of Urology, University College London Hospitals NHS Foundation Trust, London, UK

**Contributors** The authors' contribution includes, but is not limited to, the following: MT and SM wrote the manuscript. YZ and JMN created the study concept. YZ, BSSS, AF, AK, DK, HCW and ME provided supervision and guidance. All authors reviewed and approved the manuscript in its current form. MT and SM are the guarantors of this work.

**Funding** Funding for the Open Access publication charges for this article was provided by the Imperial Open Access Fund. JMN is funded by the Medical Research Council (MRC). BSSS is funded by the Royal Marsden Cancer Charity. ME receives research support from the UK's National Institute of Health Research (NIHR) UCLH/UCL Biochemical Research Centre.

**Competing interests** JMN receives funding from the MRC. BSSS receives funding from the Royal Marsden Cancer Charity. HCW receives funding from Prostate Cancer UK, the Urology Foundation and Rosetrees Trust. AK, AF and ME have stock interest in Nuada Medical. ME acts as a consultant, trainer and proctor to Sonatherm; Angiodynamics; Exact Imaging.

**Patient and public involvement** Patients and/or the public were not involved in the design, or conduct, or reporting, or dissemination plans of this research.

**Patient consent for publication** Not applicable.

**Provenance and peer review** Not commissioned; externally peer reviewed.

**ORCID iDs**
Maya Thomas http://orcid.org/0000-0001-6997-9438
Benjamin Scott S Simpson http://orcid.org/0000-0003-3685-6110
Daniel Kelly http://orcid.org/0000-0002-1847-0655
Yi Zhao http://orcid.org/0000-0002-4563-4344
Mark Emberton http://orcid.org/0000-0003-4230-0338
Joseph M Norris http://orcid.org/0000-0003-2294-0303

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
