## [Reviewer comments · BMJ Open]

ARTICLE DETAILS

TITLE (PROVISIONAL)	The use of artificial intelligence in the detection of primary prostate cancer in multiparametric magnetic resonance imaging with its clinical outcomes: a protocol for a systematic review and meta-analysis
AUTHORS	Thomas, Maya; Murali, Sanjana; Simpson, Benjamin Scott; Freeman, Alex; Kirkham, Alex; Kelly, Daniel; Whitaker, Hayley; Zhao, Yi; Emberton, Mark; Norris, Joseph

VERSION 1 – REVIEW

REVIEWER	Rouvière, Olivier Université de Lyon
REVIEW RETURNED	10-Apr-2023

GENERAL COMMENTS	The manuscript is a protocol for a systematic review and meta-analysis of studies reporting on the accuracy of artificial intelligence (AI)-based algorithms aimed at diagnosing prostate cancer on MRI. The protocol is clearly written. However, one would have expected much more details on search criteria and on inclusion/exclusion criteria in a context of highly heterogeneous literature. Here are my detailed comments: 1. Strengths and limitations of this study: “The study will be the first systematic review and meta-analysis to look solely at AI algorithms for mpMRI and see how they compare to expert evaluation, written in line with the PRISMA guidelines”. I disagree with this claim. Many systematic reviews have already been published on AI-based algorithms for prostate MRI. A quick search on PROSPERO found at least three other systematic reviews on this topic (CRD42020180083, CRD42022352281, CRD42021234054) of which, at least one has just been published. Other systematic reviews have also been published in the past years (e.g., Syer T et al, Cancers 2021; 13:3318, Sushentsev N et al, Insights Imaging 2022; 13:59). 2. It is unclear how the authors defined ‘artificial intelligence’. From their search strategy, it seems that only deep learning-based algorithms will be taken into consideration. Will this be the case? Will papers reporting on more traditional machine learning techniques, or even on quantitative MRI be excluded? This should be clearly specified and discussed.
---

	3. If only deep learning-based algorithms are considered, it may save time to reduce the search period and to start at a much later date than 1977. 4. From the systematic reviews already published, it appears that there is major heterogeneity across studies in terms of patient population (patients with proven cancer vs patients with suspected cancer), MRI protocol (biparametric vs multiparametric), human reading scores, or histological standard of reference (prostatectomy specimens vs various biopsy protocols). It is unclear whether all these studies will be accepted and what will be the exclusion criteria. If the protocol is to be published, I think it is important that a clear list of inclusion and exclusion criteria is disclosed. 5. To my knowledge, none of the systematic reviews published on the subject so far included meta-analyses. The authors renounced performing a meta-analysis given the literature heterogeneity. Could the authors comment on why they believe they would be able to perform a meaningful meta-analysis? 6. Lines 119-120: "Furthermore, variations of the ...in use". I strongly disagree with this sentence. Of course, some of the selected studies will use PI-RADS version 1 and others PI-RADS version 2 or 2.1 depending on their publication year. This is another source of heterogeneity the authors will face. However, the authors' sentence implies that both versions are currently in use, which is not true. PI-RADS version 1 is clearly outdated. Please reformulate.
--	---

REVIEWER	Sunogrot, Mohammed Norwegian University of Science and Technology, Department of Circulation and Medical Imaging
REVIEW RETURNED	15-May-2023

GENERAL COMMENTS	Overall, the protocol is well written. However, I have some minor comments that can further improve the manuscript: Line 174, Table 1: For each item, it would be helpful to know if there are predefined categories to choose from or if the collected data will be grouped into limited categories later on. For example, under "Experimental design," will the data be categorized into a specific design group before further processing? Additionally, it would be valuable to consider the type of data used for training/testing, such as whether it was acquired at 1.5/3 T or with an endorectal coil. Moreover, it would be informative to know if the model's performance was compared against the PI-RADS scores or the biopsy findings. Furthermore, it would be worth mentioning which version of the PI-RADS guidelines was followed. Line 188: Please establish a clear order for the calculations or data requests. Should the values be calculated from the tables or requested from the authors first? Alternatively, if the order can vary depending on the case, it would be beneficial to explicitly state so and provide guidelines for determining the appropriate order. Line 210: What would be the course of action in the event of a disagreement between the two reviewers? It would be helpful to
---

	outline a resolution process or provide guidelines for handling such situations to ensure consistency and minimize bias. Line 257, Draft search strategy: Consider expanding the terms used in the search strategy to include relevant terms like "machine learning" and "radiomics." This would help ensure a comprehensive literature review and capture relevant studies in the field.
--	---

VERSION 1 – AUTHOR RESPONSE

Reviewer: 1

Dr. Olivier Rouvière, Université de Lyon

Comments to the Author:

The manuscript is a protocol for a systematic review and meta-analysis of studies reporting on the accuracy of artificial intelligence (AI)-based algorithms aimed at diagnosing prostate cancer on MRI.

The protocol is clearly written. However, one would have expected much more details on search criteria and on inclusion/exclusion criteria in a context of highly heterogeneous literature.

Here are my detailed comments:

1. Strengths and limitations of this study: "The study will be the first systematic review and meta-analysis to look solely at AI algorithms for mpMRI and see how they compare to expert evaluation, written in line with the PRISMA guidelines".

I disagree with this claim. Many systematic reviews have already been published on AI-based algorithms for prostate MRI. A quick search on PROSPERO found at least three other systematic reviews on this topic (CRD42020180083, CRD42022352281, CRD42021234054) of which, at least one has just been published. Other systematic reviews have also been published in the past years (e.g., Syer T et al, *Cancers* 2021; 13:3318, Sushentsev N et al, *Insights Imaging* 2022; 13:59).

We have explored each of the suggested projects:

CRD42020180083 – The authors did not specify which aspect of prostate cancer the study is focusing on. Our review specifically focuses on the use of deep learning in mpMRI for prostate cancer diagnosis.

CRD42022352281 – This has been published. We note that the author did not include a meta-analysis due to the heterogeneity of results. We are unable to identify what could be the sources of heterogeneity proposed by the authors. We aim to explore the potential sources of heterogeneity as guided by the Cochrane Handbook of Systematic Reviews of

Interventions <https://training.cochrane.org/handbook/current/chapter-10>. Depending on the data collected, we will conduct subgroup analysis to assess the impact of specific study methodologies (type of DL used, mpMRI machine brands, definition of csPCa) on overall heterogeneity. This will assist future studies in their study development and identify potential collaboration with similar datasets for external validation.

CRD42021234054 – The author's method of data analysis includes sensitivity and specificity, accuracy. Our methods extend beyond the conventional pooled analysis to AI-specific matrices such as F1-score and Precision-Recall AUC to address the limitation of using sensitivity, specificity, and accuracy, such as the impact of class imbalance in the training dataset.

Syer T et al, *Cancers* 2021 – This author did not conduct a meta-analysis due to the potential heterogeneities in the computer-aided diagnosis (CAD) system but rather on their included studies.

However, there are several existing meta-analyses published for the use of AI in diagnostic test accuracy studies for pooled analysis of their performances:

<https://www.nature.com/articles/s41746-021-00438-z>

<https://onlinelibrary.wiley.com/doi/full/10.1111/apt.16778>

[https://www.thelancet.com/journals/landig/article/PIIS2589-7500\(19\)30123-2/fulltext](https://www.thelancet.com/journals/landig/article/PIIS2589-7500(19)30123-2/fulltext)

<https://onlinelibrary.wiley.com/doi/full/10.1111/apt.16778>

Sushentsev N et al, Insights Imaging 2022; 13:59 – The authors claimed that there are substantial heterogeneity of study characteristics which prevented a meta-analysis, but we did not find any investigation into the degree of heterogeneities and its impact on the study designs.

2. It is unclear how the authors defined 'artificial intelligence'. From their search strategy, it seems that only deep learning-based algorithms will be taken into consideration. Will this be the case? Will papers reporting on more traditional machine learning techniques, or even on quantitative MRI be excluded? This should be clearly specified and discussed.

This paper will look at all forms of artificial intelligence, including fully automated and semi-automated methodologies. The nature of both approaches means that both deep learning algorithms and more traditional forms of AI, such as machine learning methods, will be analysed. We have updated the search strategy in the supplementary file.

3. If only deep may save time to reduce the search period and to start at a much later learning-based algorithms are considered, it date than 1977.

We have reduced our search period to begin in January 2016, in line with the existing literature. This is reflected in lines 39 and 147.

4. From the systematic reviews already published, it appears that there is major heterogeneity across studies in terms of patient population (patients with proven cancer vs patients with suspected cancer), MRI protocol (biparametric vs multiparametric), human reading scores, or histological standard of reference (prostatectomy specimens vs various biopsy protocols). It is unclear whether all these studies will be accepted and what will be the exclusion criteria. If the protocol is to be published, I think it is important that a clear list of inclusion and exclusion criteria is disclosed.

The inclusion and exclusion criteria have now been amended to increase the specificity. This is reflected in lines 166-175.

5. To my knowledge, none of the systematic reviews published on the subject so far included meta-analyses. The authors renounced performing a meta-analysis given the literature heterogeneity. Could the authors comment on why they believe they would be able to perform a meaningful meta-analysis?

We wish to refer to point 1.

Through a meta-analysis, we wish to explore areas of larger heterogeneities of included studies to inform future studies in their study design better. If the heterogeneity is significant, we will discuss this in our review and give recommendations accordingly. Furthermore, subgroup analysis will enable a more profound investigation of the possible heterogeneity sources in included studies.

6. Lines 119-120: "Furthermore, variations of the ...in use". I strongly disagree with this sentence. Of course, some of the selected studies will use PI-RADS version 1 and others PI-RADS version 2 or 2.1 depending on their publication year. This is another source of heterogeneity the authors will face. However, the authors' sentence implies that both versions are currently in use, which is not true. PI-RADS version 1 is clearly outdated. Please reformulate.

We have amended and highlighted this sentence to describe the change in PI-RADS versions over time.

This is reflected in lines 125-126.

=====
Reviewer: 2

Dr. Mohammed Sunoqrot, Norwegian University of Science and Technology

Comments to the Author:

Overall, the protocol is well written. However, I have some minor comments that can further improve the manuscript:

Line 174, Table 1: For each item, it would be helpful to know if there are predefined categories to choose from or if the collected data will be grouped into limited categories later on. For example, under "Experimental design," will the data be categorized into a specific design group before further processing? Additionally, it would be valuable to consider the type of data used for training/testing, such as whether it was acquired at 1.5/3 T or with an endorectal coil. Moreover, it would be informative to know if the model's performance was compared against the PI-RADS scores or the biopsy findings. Furthermore, it would be worth mentioning which version of the PI-RADS guidelines was followed.

We plan to collect data without predefined categories and group data later after reviewing the extracted studies. We have added the type of PI-RADS guideline used in Table 1 and will consider the type of MRI imaging used.

Line 188: Please establish a clear order for the calculations or data requests. Should the values be calculated from the tables or requested from the authors first? Alternatively, if the order can vary depending on the case, it would be beneficial to explicitly state so and provide guidelines for determining the appropriate order.

We have now explicitly stated the order in which we will go about dealing with data requests. If the reported true positive, false negative, true negative and false positive figures were not available, attempts will be made to calculate the required data ourselves (sensitivity and specificity), but if this should fail, then the authors will be contacted to provide any missing data.

This is reflected in lines 182-185.

Line 210: What would be the course of action in the event of a disagreement between the two reviewers? It would be helpful to outline a resolution process or provide guidelines for handling such situations to ensure consistency and minimize bias.

In the event of disagreement between two reviewers, a third reviewer will be consulted, and the issue will be resolved by consensus. This has been highlighted in our study selection and data extraction methodology.

This is reflected in lines 160-162 and 179-181.

Line 257, Draft search strategy: Consider expanding the terms used in the search strategy to include relevant terms like "machine learning" and "radiomics." This would help ensure a comprehensive literature review and capture relevant studies in the field.

We have expanded the search strategy and included the updated version in a supplementary file per the editor's comments.

Reviewer: 1

Competing interests of Reviewer: None

Reviewer: 2

Competing interests of Reviewer: I have no competing interests.

VERSION 2 – REVIEW

REVIEWER	Sunoqrot, Mohammed Norwegian University of Science and Technology, Department of Circulation and Medical Imaging
REVIEW RETURNED	13-Jul-2023
GENERAL COMMENTS	The comments and suggestions were well addressed.